# The Effect of Formulation Variables on the Manufacturability of Clopidogrel Tablets via Fluidized Hot-Melt Granulation—From the Lab Scale to the Pilot Scale

**DOI:** 10.3390/pharmaceutics16030391

**Published:** 2024-03-13

**Authors:** Béla Kovács, Erzsébet-Orsolya Tőkés, Éva Katalin Kelemen, Katalin Zöldi, Francisc Boda, Edit Suba, Boglárka Kovács-Deák, Tibor Casian

**Affiliations:** 1Department F1, Biochemistry and Environmental Chemistry, George Emil Palade University of Medicine, Pharmacy, Science and Technology of Târgu Mureș, 540142 Târgu Mureș, Romania; bela.kovacs@umfst.ro; 2Gedeon Richter Romania, 540306 Târgu Mureș, Romania or tokes.erzsebet-orsolya.23@stud.umfst.ro (E.-O.T.); katalin.kelemen@gedeon-richter.ro (É.K.K.); katalin.zoldi@gedeon-richter.ro (K.Z.); edit.suba@gedeon-richter.ro (E.S.); 3The Doctoral School of Medicine and Pharmacy, Institution Organizing University Doctoral Studies, George Emil Palade University of Medicine, Pharmacy, Science and Technology of Târgu Mureș, 540142 Târgu Mureș, Romania; bogyo131@gmail.com; 4Department F1, General and Inorganic Chemistry, George Emil Palade University of Medicine, Pharmacy, Science and Technology of Târgu Mureș, 540142 Târgu Mureș, Romania; 5Department of Pharmaceutical Technology and Biopharmacy, “Iuliu Hațieganu” University of Medicine and Pharmacy, 400012 Cluj-Napoca, Romania; casian.tibor@umfcluj.ro

**Keywords:** design of experiments, multivariate data analysis, fluidized hot-melt granulation, pharmaceutical process scale-up, pharmaceutical development

## Abstract

Solid pharmaceutical formulations with class II active pharmaceutical ingredients (APIs) face dissolution challenges due to limited solubility, affecting in vivo behavior. Robust computational tools, via data mining, offer valuable insights into product performance, complementing traditional methods and aiding in scale-up decisions. This study utilizes the design of experiments (DoE) to understand fluidized hot-melt granulation manufacturing technology. Exploratory data analysis (MVDA) highlights similarities and differences in tablet manufacturability and dissolution profiles at both the lab and pilot scales. The study sought to gain insights into the application of multivariate data analysis by identifying variations among batches produced at different manufacturing scales for this technology. DoE and MVDA findings show that the granulation temperature, time, and Macrogol type significantly impact product performance. These factors, by influencing particle size distribution, become key predictors of product quality attributes such as resistance to crushing, disintegration time, and early-stage API dissolution in the profile. Software-aided data mining, with its multivariate and versatile nature, complements the empirical approach, which is reliant on trial and error during product scale-up.

## 1. Introduction

In 2002, the United States Food and Drug Administration (FDA) introduced the Pharmaceutical cGMPs for the 21st Century—a Risk-Based Approach, a guideline that moved the focus from Quality by Testing (QbT) to Quality by Design (QbD) during the manufacturing of pharmaceutical products [1]. QbD highlights all possible sources of variability during production, offering solutions to mitigate variations and maintain the desired quality of the final product. By modeling and predicting correlations between variables and end results, QbD leads to reduced experimentation and, consequently, to lower production and analysis costs [2]. As the manufacturing of pharmaceutical products includes many phases, all with their own variables, large datasets become available. These data can be efficiently processed only through computational methods, mainly through multivariate data analysis (MVDA), an integral part of the QbD approach. MVDA represents a data analysis method in which the correlation between more than one independent variable (input) and/or dependent variables (output) is evaluated. This allows the identification of underlying correlations, trends, and patterns that would be difficult or impossible to observe without computational assistance [3,4,5,6]. It is an integral part of the Process Analytical Technology (PAT) methodology, which aims to improve the understanding, optimization, monitoring, and control of various processes involved in the manufacturing of pharmaceutical products [7]. Moreover, MVDA proves to be a powerful tool in the modeling and prediction of end products, being an important asset in granulation techniques [8,9,10,11], various pharmaceutical applications [12,13,14,15], and the scale-up process (from the laboratory scale to the pilot scale or the pilot scale to the industrial scale) during production development [16,17,18].

Fluidized hot-melt granulation (FHMG) is a solvent-free pharmaceutical manufacturing technology that implies the use of low-melting-point binders (50–80 °C), carrying out granulation with hot air as a heat source. This technique can be advantageous when processing water-sensitive active pharmaceutical ingredients, and it benefits from a reduced processing time [19,20,21].

Pharmaceutical process scale-up often represents a herculean task for technologists. Typically, during process scale-up, three types of similarities should be considered between equipment: geometric, kinematic, and dynamic [22]. The scale-up process from the laboratory scale to the pilot scale in FHMG presents a unique set of challenges and opportunities that significantly affect the successful translation of a formulation concept into a larger manufacturing environment. This innovative granulation technique, involving the use of a fluidized bed to granulate and coat particles with a molten binder, offers several advantages, but its transition to the pilot scale demands a careful consideration of various factors. FHMG relies on the precise control of heat and mass transfer. As the scale increases, challenges may arise in maintaining uniform temperature profiles and ensuring consistent particle growth, impacting the overall product quality. Transitioning from laboratory-scale equipment to pilot-scale machinery requires a careful consideration of design modifications. A successful scale-up requires a holistic approach, integrating process optimization, equipment design considerations, and a thorough understanding of material behavior. The most relevant input variables for FHMG are binder content and binder particle size, inlet air temperature and/or flow rate, granulation time, and product endpoint temperature [19,23,24,25]. Monitored output variables include granule size, size distribution, and growth kinetics, as well as physical and technological properties of granules (shape, porosity, flowability, and compressibility) and tablets (resistance to crushing and dissolution) [19,20,23,26].

As only a limited number of studies deal with the application of FHMG optimization, the aim of the present study was the development of an immediate-release, solid oral dosage formulation containing Clopidogrel hydrogen sulfate, form II, via FHMG in the framework of quality-by-design methodologies. The aim of the study was to obtain insights into the development and scale-up of an FHMG process using multivariate data analysis by mapping differences between batches manufactured at different manufacturing scales for this type of technology.

Marketed under the trade name Plavix 75 mg for film-coated tablets and other brands, Clopidogrel is a medication employed as an antiplatelet agent to mitigate the likelihood of heart disease and stroke in individuals deemed to be at elevated risk [27]. Clopidogrel belongs to the class II active substances as categorized by the Biopharmaceutics Classification System (BCS), showing low solubility and high permeability. This makes the active pharmaceutical ingredient an optimal candidate for formulation studies, as its pharmaceutical and in vitro performance will be confined due to the in vivo fate of tablets, taking into consideration that Clopidogrel hydrogen sulfate possesses high permeability. Among the six polymorphic forms described, form II is thermodynamically more stable, with a melting point of approx. 177 °C, having superior compaction properties compared to form I [28]. Sanofi Synthelabo SA claimed that the use of polymorphic form II in pharmaceutical formulations, as disclosed in patent no. EP1087976, expired in 2019, making generic formulations containing Clopidogrel hydrogen sulfate, form II, possible [29].

## 2. Materials and Methods

### 2.1. Materials Used for FHMG and Analytical Measurements

The Ph. Eur.-grade active pharmaceutical ingredient Clopidogrel hydrogen sulfate, form II, was obtained from MSN Laboratories Ltd. (Telangana, India). The excipients used for FHMG were Mannitol 35 (Roquette Frères, Lestrem, France) and cellulose, microcrystalline type 103D+ (Mingtai Chemical Co., Ltd., Taoyuan City, Taiwan) as fillers, Macrogol 6000 or Macrogol 8000 (Dow Chemical Company, Hahnville, LA, USA) as plasticizers, and low-substituted hydroxypropyl-cellulose L-HPC, LH-11 (Shin-Etsu Chemical Co., Ltd., Tokyo, Japan), as a binder. Hydrogenated castor oil, Kolliwax HCO (BASF SE, Ludwigshafen, Germany), was used as a lubricant. Analytical-grade potassium chloride, potassium hydroxide, and hydrochloric acid were obtained from Merck (Merck GmbH, Darmstadt, Germany) and used for the preparation of dissolution media.

### 2.2. Manufacturing Technology

#### 2.2.1. Pharmaceutical Formula

The qualitative and quantitative formulas of the tablets are listed in Table 1.

#### 2.2.2. Technological Process

Mannitol 35 and the selected type of Macrogol were sieved through a Ø = 1.0 mm pore size. Next to the sieved material, Clopidogrel, L-HPC, and cellulose, microcrystalline type 103D+ was added. This powder mass was subjected to FHMG in a fluid-bed granulator–dryer. The obtained granules were sieved and lubrified with Kolliwax HCO. The obtained lubricated granules were compressed using round, biconcave, Ø = 8.0 mm punches. A flowchart of the applied manufacturing technology is presented in Figure 1.

#### 2.2.3. Equipment

The equipment used for the manufacturing of Clopidogrel tablets at the laboratory scale was as follows:Frewitt-Coniwitt Lab rotary sieve with a Ø = 1.0 mm sieve insert (Frewitt Ltd., Fribourg, Switzerland);Bosch Solidlab 1 fluid-bed granulator and dryer (Bosch GmbH, Schopfheim, Germany);Erweka AR402 double-cone blender (Erweka GmbH, Langen, Germany);RIVA Piccola D8 rotary tablet press (RIVA S.A., Ciudadela, Buenos Aires, Argentina);Bosch Solidlab 1 film-coating machine (Bosch GmbH, Schopfheim, Germany).

Experiments on the pilot scale were carried out using the following equipment:Glatt GS-100 rotary sieve with a Ø = 1.0 mm sieve insert (Glatt GmbH, Binzen, Germany);Hüttlin Pilotlab fluid-bed granulator and dryer (Hüttlin GmbH, Schopfheim, Germany);Mini Kiskun-Meridián drum blender (Kiskun Meridián Ipari Gépgyártó Zrt., Kiskunfélegyháza, Hungary);Korsch XL 100 (Korsch AG, Berlin, Germany);GS-HP/F 025 film-coating machine (I.M.A. Industria Macchine Automatiche S.P.A., Ozzano Dell’Emilia (BO), Italy).

### 2.3. Product and Process Optimization at the Laboratory Scale

#### 2.3.1. Risk Assessment

Preliminary hazard analysis based on empirical consideration, complemented with an Ishikawa diagram and a subsequent FMECA methodology, was applied for the definition of putative critical process parameters [30].

#### 2.3.2. Experimental Design for Critical Process Parameters (CPP) and Critical Material Attributes (CMA) Screening

A screening, full-factorial experimental design was conducted to find optimal setpoints of the CPPs and analyze the impact of CMAs that, by means of the performed risk analysis, might influence, in a significant manner, product performance. The design of experiments and the analysis of raw data were achieved using the MODDE 13 software (Sartorius Stedim Data Analytics AB, Umeå, Sweden). The factors defined and responses selected are presented in Table 2. Factors were chosen to obtain further information about the developed fluidized hot-melt granulation. Having taken into consideration that the melting points of Macrogol 6000 and Macrogol 8000 are between 55–63 °C and 60–63 °C, respectively, we aimed to investigate the granulating potential of these polyethylene glycols at lower and higher temperatures (50 °C to 65 °C, below and above the melting temperature, LX1), for a specified granulation time (5 to 30 min, LX2). The different Macrogol types were chosen based on their applicability in a fluidized hot-melt granulation process and according to the Functionality-Related Characteristics defined in the European Pharmacopoeia [31]. In this regard, two different types of Macrogols were chosen, based on their differences in viscosity (Macrogol 6000 vs. Macrogol 8000) and differences in particle size (Macrogol 8000, powder grade vs. Macrogol 8000, coarse grade, LX3). The selected responses were chosen for characterization via the in-process control of the obtained granules, measuring the bulk density (LY1), flowability (LY2), and granulometric distribution (LY3a–LY3f). The experiments conducted at the laboratory scale within the experimental design are detailed in Appendix A, comprising batches L1–L15. These trials aimed to accumulate insights into the FHMG process; therefore, the laboratory-scale batches were not lubricated and compressed into tablets.

#### 2.3.3. Effect of the Lubricant Level on Clopidogrel Dissolution

The effect of the lubricant on Clopidogrel dissolution was also assessed at the laboratory scale. Batches with 1.33% (initial), 1.67% (+25% increase), and 2.00% (+50% increase) Kolliwax HCO content were manufactured for this purpose. The influence of the lubricant was investigated for both Macrogol types. Batches L18 and L21 were manufactured with 1.33% (*w*/*w*) Kolliwax HCO, incorporating PEG 8000P and PEG 6000P, respectively. Batches L16 and L17 were manufactured using 1.67% (*w*/*w*) Kolliwax HCO, incorporating PEG 8000P and PEG 6000P, respectively. Batch no. L19 was manufactured using 2.00% (*w*/*w*) Kolliwax HCO, incorporating PEG 8000P as a binder. The lubrication step was carried out in a double-cone blender for two minutes at 10 rpm.

### 2.4. Product and Process Optimization at the Pilot Scale

The product scale-up to the pilot scale was conducted based on the information obtained at the laboratory scale. In general, the inlet airflow and equipment loading in terms of the unit of air volume per kg of the granule bed per minute (m^3^/kg/min) and the inlet air temperature were kept as similar as possible. Furthermore, we aimed to investigate the manufacturability of Clopidogrel tablets with PEG6000 and PEG8000 at the pilot scale as well. The tablet compression step was scaled up based on tablet in-process control characteristics, but taking into consideration the differences in equipment (RIVA vs. Korsch), we applied a separate experimental design for the elucidation of the most influential factors of tablet compression.

#### Experimental Design for Tablet Compression

A separate screening experimental design was applied on the pilot scale in order to elaborate upon the CPPs for tablet compression. The factors considered were compression force (PX1), turret rotation speed (PX2), and feeder rotation speed (PX3). The impact of tablet compression variables on the manufacturability of tablet cores was evaluated by registering the disintegration time (PY1), Cpk (PY2), resistance to crushing (PY3), and friability (PY4) of these cores (Table 3).

### 2.5. Product Characterization

#### 2.5.1. In-Process Control of Granules and Tablets

The granule flow-out time and bulk density were determined using a Pharma Test PTG-S4 powder testing system (Pharma Test Apparatebau AG, Hainburg, Germany). Granulometric distribution was determined with a vibratory sieve shaker, the Analysette 3 PRO (Fritsch GmbH, Weimar, Germany). The loss on drying of the granules was measured using a Mettler Toledo HR-73-type halogen moisture analyzer (Mettler Toledo, Columbus, OH, USA) using a standard drying method of 70 °C for 20 min. Tablet physical testing was performed using a Pharmatest WHT 3ME automated tablet testing system (Pharma Test Apparatebau AG, Hainburg, Germany). The disintegration time was determined using a PTZ Auto 1 Single Position Semi-Automated Disintegration Testing Instrument (Pharma Test Apparatebau AG, Hainburg, Germany). Tablet friability was assessed with a Pharmatest PTF 10E friability tester (Pharma Test Apparatebau AG, Hainburg, Germany). The process capability index of tablet mass variation was calculated based on the following formula using the Minitab 15 data analysis software:$$C_{pk}=min\left[\frac{USL-\mu }{3\sigma },\frac{\mu -LSL}{3\sigma }\right]$$
where *μ* is the mean weight, *σ* is the standard deviation of 20 tablet cores, and *USL* and *LSL* are the upper and lower specification limits of ±7.5% according to the 2.9.5 monograph of the European Pharmacopoeia.

#### 2.5.2. In Vitro Release Study of Film-Coated Tablets

Dissolution studies of laboratory-scale batches L16–L25 and pilot-scale batches P1–P7 were performed by means of an Erweka DT800 automated dissolution tester system (Erweka GmbH, Langen, Germany). Clopidogrel dissolutions were recorded using 1000 mL of a KCl/HCl buffer system at pH = 2.0 ± 0.05 as dissolution media, setting a 50 rpm paddle rotation speed and a temperature of 37 ± 0.5 °C, as presented in the United States Pharmacopoeia monograph for Clopidogrel tablets [32]. Sampling was carried out at 5, 10, 15, 20, 30, and 45 min. Analytical signal detection was carried out at λ = 240 nm using a Shimadzu UV-1800 UV/Visible Scanning Spectrophotometer (Shimadzu Corporation, Kyoto, Japan). A comparison of dissolution profiles was evaluated by calculating the similarity factor, *f*2, between the sets of results using the following equation:$$f2=50\;{log}_{10}\;\left\{{\left[1+\frac{1}{n}\sum_{t=1}^{n}w_{t}{\left(R_{t}-T_{t}\right)}^{2}\right]}^{-0.5}\times 100\right\},$$
where *n* is the number of dissolution points, *R_t_* and *T_t_* are the dissolution values of the reference and test product at time *t*, respectively, and *w_t_* is an optional weighting factor.

### 2.6. Multivariate Data Analysis

OPLS-DA (Orthogonal Projections to Latent Structures-Based Discriminant Analysis) models were developed to investigate the differences between batches prepared at the laboratory and pilot scales. Variables describing processing conditions (granulation temperature and compression force), formulation composition (Macrogol type and Kolliwax content), granule quality attributes (bulk density, flow-out time, and size fractions), and tablet characteristics (disintegration time, resistance to crushing, and dissolution profile) were scaled to the unit variance. The model performance was evaluated by considering the predictive capacity (Q2) and the percentage of the explained variability (R2X and R2Y). Variables with discriminatory power between the two classes were identified by generating loading column plots.

OPLS models were developed to highlight the differences in the effect of processing conditions at different manufacturing scales. In this stage of data analysis, the Y variables were represented by tablet critical quality attributes with a discriminatory power in the OPLS-DA models. The comparison of loading coefficients in terms of direction and magnitude enabled an effective comparison of manufacturing scales. Data analysis was performed with SIMCA 17 (Sartorius Stedim Biotech, Goettingen, Germany). For the evaluated database, see Appendix A.

## 3. Results and Discussion

### 3.1. Risk Assessment

Fluidized hot-melt granulation (FHMG) represented the most critical step in the manufacturing of Clopidogrel 75 mg film-coated tablets. The bed temperature during FHMG was critical due to the melting points of Macrogol 6000 and Macrogol 8000, 55–63 °C and 60–63 °C, respectively. In addition, the time of granulation represented a hazard; if a shorter time was achieved, the granulation would have been incomplete, resulting in compression difficulties. Furthermore, an excessive granulation time was avoidable to prevent the formation of material conglomerates [33,34,35].

Lubrication was handled as a critical step, given its importance in granule in-process control (flowability) and amending the tablet compression process by reducing the stickiness of the granulated material. Compression force and turret/feeder rotation speed were considered critical parameters in tablet compression, affecting material behavior and tablet in-process control results [36].

Walker et al. have similarly shown that binder content and viscosity, particle size distribution, and granulation time are key CMAs and CPPs for the tablet pressing of pharmaceutical powders manufactured via FHMG [21]. Our findings show that, in this particular setting, the temperature of the fluidized bed (LX1) and the granulation temperature (LX2) are the most relevant parameter settings that would influence the considered responses, whilst granule growth, especially in the range above 1.0 mm, is influenced by the type of the incorporated PEG. A detailed Ishikawa Chart is presented in Appendix A.

### 3.2. Experimental Design for CPP and CMA Screening

Of the 15 batches manufactured for CPP and CMA screening purposes, L1 resulted in batch failure due to impossible tablet compression, and it was excluded from design evaluation. The loss on drying of the batches was situated between 0.17–0.53%, having a bulk density ranging from 0.523–0.595 g mL^−1^. In terms of the particle size distribution, a clear distinguishment in granulometric patterns can be observed between the manufactured batches (Appendix A).

Apart from bulk density, all PLS models presented acceptable goodness of fit *R*^2^ > 0.5 and goodness of predictability *Q*^2^ > 0.25 values (Appendix A), and the regression models were considered statistically significant (*p* < 0.01).

Bulk density (LY1) was not influenced by any of the tested factors (*p* > 0.05); the variability of results ranged only from 0.523 g mL^−1^ to 0.595 g mL^−1^, indicating that no important response variation was captured among the different manufacturing set points.

The flow-out time (LY2) presented a strong negative correlation (*p* < 0.001) with the temperature of the fluidized bed (LX1) during FHMG (Figure 2a). This is in concordance with the melting temperature of the Macrogol in the composition, as the fluidized bed must be heated to around 60–65 °C in order to obtain proper granulation. The negative correlation between the flow-out time and the fluidized bed temperature implies that controlling and optimizing the temperature during FHMG was crucial to achieving the desired granulation outcomes. It also suggested that deviations from the optimal temperature, especially below the melting point of Macrogol, may lead to suboptimal granulation, affecting the flow-out time.

Particle size distribution in the lower span of <200 μm (LY3e and LY3f) showed an inverse proportionality with the granulation temperature (LX1) and the granulation time (LX2). The granulation temperature showed a direct, proportional factor-to-response relationship and mainly influenced the particle size greater than 200 μm (Figure 2b). In addition to the granulation temperature, excessive granule growth, i.e., particles greater than 1 mm, was greatly influenced by the type of Macrogol incorporated in the pharmaceutical formulation. Whilst powder-grade PEGs reduced the quantity of such large particles, coarse-grade-type PEGs favored granule growth. The optimal granulation temperature resulted in an appropriate PSD and flow-out time simultaneously.

By analyzing laboratory-scale batches manufactured with different types of Macrogol but with identical factor settings (65 °C, 10 min, and 1.33% lubricant; L18 with Macrogol 8000P vs. L25 with Macrogol 6000P), it was revealed that powder-grade Macrogol 6000P accelerates the Clopidogrel release compared to powder-grade Macrogol 8000P, although this was not a drastic impact. Intriguingly, different types of Macrogol favored granule growth in different spans of the particle size distribution. Macrogol 6000P contributed to a higher proportion of granules in the range of 800–1000 μm, whereas Macrogol 8000P increased particle growth in the span of 200–400 μm (Figure 3).

### 3.3. Effect of the Lubricant Level on Clopidogrel Dissolution

The generated bi-plot (Figure 4a) of the PCA-X modeling of these batches indicated that the quantity of Kolliwax HCO incorporated in the pharmaceutical formulation mainly influenced the resistance to crushing of the tablet cores. Batches L18 and L21, manufactured with the lowest Kolliwax HCO content (1.33%), exhibited the fastest dissolution, and differences in the Macrogol types were once again evident. Batch no. L18, manufactured with PEG8000P, showed a slower release of Clopidogrel compared to L21 manufactured with PEG6000P. On the other end, batch no. L19, manufactured with 2.00% Kolliwax HCO, showed the slowest dissolution, as not only did PEG8000P hinder the release of Clopidogrel, but the effect of the lubricant on core and film-coated tablet disintegration was also evident.

When comparing batches with an identical PEG type (8000P) and different Kolliwax HCO content (L18 vs. L19), the difference in disintegration time was elucidated using the Score Contribution Plot that was obtained (Figure 4b). This increase in disintegration time corroborated well with the hindering of Clopidogrel dissolution for these formulations. The similarity factor, *f*2, results decreased accordingly, resulting in unsatisfactory values even at a 25% increase in the lubricant content.

Kolliwax HCO, due to its barrier-creating properties, might alter the matrix of the tablet, impeding rapid breakdown and resulting in slower disintegration. This delayed disintegration, in turn, correlated with the hindered dissolution of Clopidogrel (Figure 5a). Despite the influence on the disintegration time and dissolution, the quantity of Kolliwax HCO did not influence the resistance to the crushing of tablets.

In practical terms, these findings implied that the careful consideration and optimization of the amount of Kolliwax HCO in the formulation were necessary to achieve the desired disintegration characteristics and the dissolution profile of Clopidogrel. Balancing these factors is essential to ensure the effectiveness and reliability of the pharmaceutical product in terms of both formulation integrity and therapeutic outcomes.

### 3.4. Scale-Up from the Laboratory Scale to the Pilot Scale

The initial pilot-scale experiments (P1 and P2) employed PEG6000P, with the fluidized bed heated to 65 °C and maintained for 10 min. After the comparison of batches L21–L23 and L25 with the P2.4 batch, it became evident that, despite minimal differences in the in-process control results with identical pilot-scale parameter settings, achieving a similar Clopidogrel release required only adjusting the compression force (Figure 6a). Subsequently, batches P3 and P4 were produced at a lower granulation temperature (60 °C) using PEG8000P and PEG6000P, respectively, to assess the impact of plasticizers at the pilot scale as well. Finally, pilot-scale batches P5–P7 were manufactured with the same process parameter settings to verify the inter-batch variability of the manufacturing process. By comparing these batches to their laboratory-scale counterparts (L18 and L24), it can be observed that, by reducing the granulation temperature, despite a higher compression force, a higher release of Clopidogrel can be achieved in the later time points, whilst Clopidogrel dissolution is hindered at the earlier sampling points (Figure 6b).

Discrepancies between batches produced at the laboratory scale and the pilot scale were evident in bulk density, which was attributed to variations in granulometric characteristics. Batches manufactured at the laboratory scale displayed unimodal distribution, with almost half of the granules falling within the 200–400 μm range. Conversely, batches produced at the pilot scale exhibited a bimodal distribution in which the granules were primarily concentrated in the 80–200 μm range (Appendix A).

The observed difference in PSD between the laboratory- and pilot-scale batches did not significantly affect the release of Clopidogrel from the pharmaceutical dosage form (L21 vs. P2.4). However, when the pilot-scale batches manufactured with PEG6000P at various granulation temperatures were examined, there was a minimal effect on the dissolution that resulted, with a faster release observed at the 15- and 20-min time points. Conversely, batches manufactured with PEG8000P (P3.3 and P5–7) exhibited a lower release of Clopidogrel, as anticipated (Figure 5b).

### 3.5. Experimental Design for Tablet Compression

Our aim was also to investigate the effect of compression parameters on product QTPPs (quality target product profiles).

The model performance indicators returned good results, as both goodness of fit and goodness of predictability were over the threshold of 0.5 and 0.25, respectively (Appendix A). The obtained regression models were statistically significant, with *p* < 0.05 for all defined responses. A lack of fit was observed in the case of resistance to crushing, PY3 (*p* = 0.029), driven by the high reproducibility of the obtained results.

Intriguingly, compression force (PX1) did not have a significant impact on the tablet disintegration time (PY1) within the range of 16–22 kN, yet it significantly influenced the tablet friability (PY4), suggesting that, at elevated compression loads, there is a likelihood of powder lamination, resulting in a basic form of tablet capping.

In contrast, the rotation speed of the turret (PX2) showed a significant negative correlation not only with the disintegration time (PY1) but also with the resistance to crushing of the tablets as well (PY3). This observation indicated that higher productivity yields tablets with lower resistance to crushing and, subsequently, lower disintegration times. This might be explained with reference to the reduced dwell time of punch penetration during the compression phase. Dwell time significantly affects tablet quality, impacting strength and facilitating a smooth product transfer between presses; extending it reduces air entrapment, which is crucial for maintaining tensile strength and preventing issues like delamination or capping.

The rotation speed of the feeder (PX3) crucially impacted the evenness of tabletability in terms of tablet mass variation (PY2), while the turret rotation speed (PX2) only demonstrated a tendency toward significance (*p* = 0.10). Aligning with expectations, feeder rotation speed (PX3) contributed to a uniform die filling, and it showed an inverse relationship with turret speed (PX2) concerning the process capability of tablet compression. Upon comparing the loading plots for Cpk (PY2) and resistance to crushing (PY3), it was evident that the turret rotation speed (PX2) significantly influenced the resistance to crushing, while the rotation speed of the feeder (PX3) had only a marginal impact. Conversely, in terms of Cpk, the situation was reversed, with PX3 significantly influencing tablet mass variation and PX2 being only tentatively crucial in affecting this property (Figure 7).

### 3.6. Exploratory Data Analysis

Granulation techniques, according to their nature of complexity, are non-linear when a process scale-up is considered [37]. As such, different approaches, from empirical considerations to mathematical approaches, are employed to achieve a manufacturing technology that can ensure product performance.

In our approach, the developed OPLS-DA model was able to compare and discriminate between laboratory-scale and pilot-scale batches for which dissolution data exist. The orthogonal latent variable to[1] captured the variability in the dissolution profiles, with higher scores coinciding with a slower in vitro Clopidogrel release at all time points (Figure 8a).

Variations between laboratory-scale and pilot-scale batches were identified through pq[1] loading (Figure 8b). In the laboratory-scale batches, processing typically occurred at elevated material temperatures, resulting in more distinct granule growth within the range of 200 to 1000 μm. In contrast, pilot-scale batches, produced at lower temperatures to assure similarity to the commercial product, exhibited granulometry with a more powdery texture, which implied the use of a higher compression force. Interestingly, this less-granulated material had a swifter flow-out time. Also, the pilot-scale batches had a greater moisture content and bulk density, in contrast to the laboratory-scale batches. The increase in compression force yielded a higher resistance to crushing for both tablet cores and film-coated tablets, although this did not have a significant impact on dissolution.

Based on these data, we aimed to create models from laboratory- and pilot-scale batches to achieve a list of predictors for product performance indicators.

#### 3.6.1. Resistance to Crushing

An intriguing observation was the apparent independence of tablet core resistance to crushing from the applied compression force (15.5–16.5 kN at the laboratory scale and 10–24 kN at the pilot scale). Tablet hardness was predominantly influenced by the granulation temperature, contributing to enhanced flowability by favoring the formation of larger granules and reducing the proportion of the powder fraction (<80 μm) in the final blend (Figure 9). By comparing laboratory- and pilot-scale batches, a noteworthy difference can be observed in terms of factors influencing the resistance to crushing. While, at the laboratory scale (Figure 9a), a more granulated material yielded a lower tablet hardness, at the pilot scale (Figure 9b), the contrary can be seen; a higher PSD resulted in firmer tablets. The decrease in compactibility might be attributed to the involvement of compaction energy in tablet formation. In the case of dense granules, a substantial portion of the compression force is dedicated to breaking up the granules and particle rearrangement, leading to a reduction in the strength of the inter-particulate bonds [38]. This could clarify the contrasting trends in resistance to crushing observed between manufacturing scales, contingent on the resulting particle patterns. The observed difference in tablet hardness between the laboratory- and pilot-scale batches highlights the complex and scale-dependent nature of tablet manufacturing. The divergent outcomes suggest that the factors influencing resistance to crushing vary between the two scales, emphasizing the importance of considering the specific conditions and parameters associated with each scale. At the laboratory scale, the tendency for a more granulated material to yield a lower tablet hardness implies that the manufacturing process at this scale responds differently to granule characteristics. This could be attributed to factors such as mixing efficiency, compaction forces, or dwell times, which may have a more pronounced impact on tablet properties when working with smaller quantities in a laboratory setting. Contrastingly, the observation at the pilot scale indicated that a higher PSD leads to the production of tablets with greater firmness. This suggests that the scale-up process introduces new dynamics or efficiencies that alter the relationship between granule characteristics and tablet hardness.

#### 3.6.2. Disintegration Time

Another important product performance parameter that could influence the release of Clopidogrel from the pharmaceutical dosage form was the disintegration time of tablet cores. This parameter was mainly dependent on the granulation temperature and PSD of the powder blend (Figure 10). As discussed in the previous section, PSD influenced the resistance to crushing of the tablet cores, which, in turn, implied variations in the disintegration time. Increasing the disintegration time was generally achieved using PEG8000P, which, as stated previously, resulted in an increased fraction of particle size in the range of 200–400 μm.

#### 3.6.3. Dissolution

Finally, understanding the impact of process parameters at different scales for complex technologies such as FHMG would be of key importance to ensure a proper release based on process parameters and in-process control predictors. Generally, in the case of rapidly dissolving pharmaceutical products, complete dissolution is achieved in 15 or 30 min; as such, the release of Clopidogrel in the first time points would confine the swiftness of its release. As depicted in Figure 11b, a notable disparity in dissolution between laboratory and pilot scales was evident solely at the initial time-point of 5 min. However, at subsequent sampling points, the release of Clopidogrel appeared quasi-similar between the two scales. This suggested that the scale-up of the manufacturing technology can be deemed successful. Figure 11a,b illustrated variations in factors affecting Clopidogrel release at the 5-min mark between laboratory and pilot scales. At both scales, the type of Macrogol integrated into the pharmaceutical formulation exerted an identical impact on the release of Clopidogrel, this influence being more pronounced at the pilot scale. Likewise, the applied compression force demonstrated an inverse relationship with the quantity of dissolved Clopidogrel in batches at both the laboratory and pilot scales. Furthermore, at the pilot scale, parameters tied to granule manufacturing—such as particle size distribution (PSD) in the range of 80–400 μm and powder density—impacted the dissolution of Clopidogrel.

## 4. Conclusions

In formulation development, the choice of lubricants can significantly impact the release of Clopidogrel. Kolliwax HCO hindered the dissolution of Clopidogrel, leading to *f*2 values near the acceptance threshold of 50 instead of considerably higher values observed with ideal set points. Scaling up FHMG processes poses challenges, with our model revealing that a lower granulation temperature is needed at the pilot scale to achieve an appropriate granulometric distribution. A more uniform, unimodal granulometric distribution at the pilot scale compared to the bimodal facet at the lab scale ensured consistent tablet compression and product performance.

Predicting QTPPs from process parameters and in-process control data is a current challenge and opportunity in the pharmaceutical industry. Our findings suggest that valuable information can be extracted from large datasets using both empirical and systematic computer-aided techniques. The results indicate that the enumerated product quality indicators are mainly dependent on the type of Macrogol used in the pharmaceutical formulation and the granulometric distribution of the obtained granules derived from manufacturing parameter settings.

Although there were limitations to our study, such as the necessity for additional exploration of set points at the pilot scale, the results we obtained can provide guidance to formulation technologists. By contemplating alternative strategies beyond empirical methods and utilizing data mining and software-aided techniques, they can enrich their understanding of products. This, in turn, ensures elevated product quality that aligns with specifications and fulfills patient requirements.

## Figures and Tables

**Figure 1 pharmaceutics-16-00391-f001:**
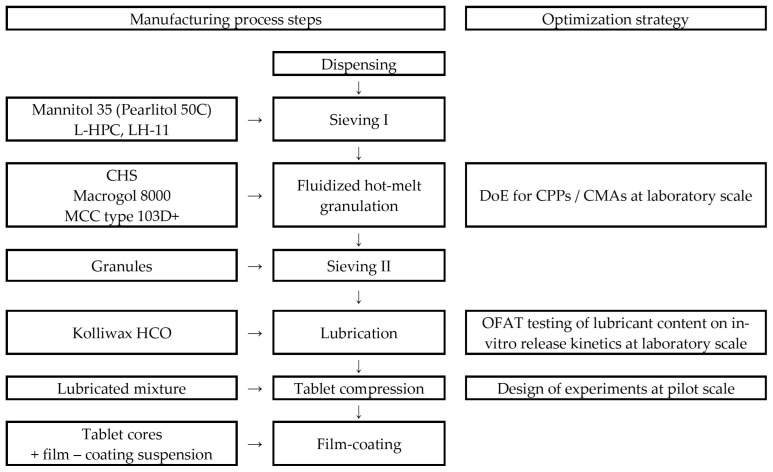
Steps of the manufacturing process for Clopidogrel tablets and presentation of the optimization strategy; OFAT—one factor at time, CPP—critical process parameter, and CMA—critical material attribute.

**Figure 2 pharmaceutics-16-00391-f002:**
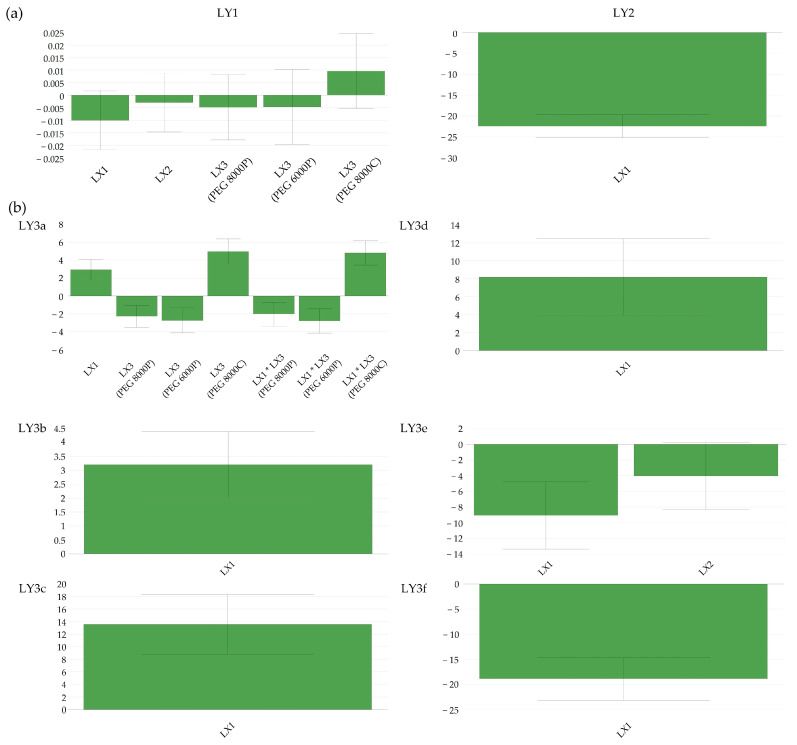
(**a**) Coefficient plots indicating the direction and magnitude of impact of defined factors on bulk density (LY1) and flow-out time (LY2). (**b**) Coefficient plots indicating the direction and magnitude of impact of defined factors on particle size distribution (LY3a–LY3f).

**Figure 3 pharmaceutics-16-00391-f003:**
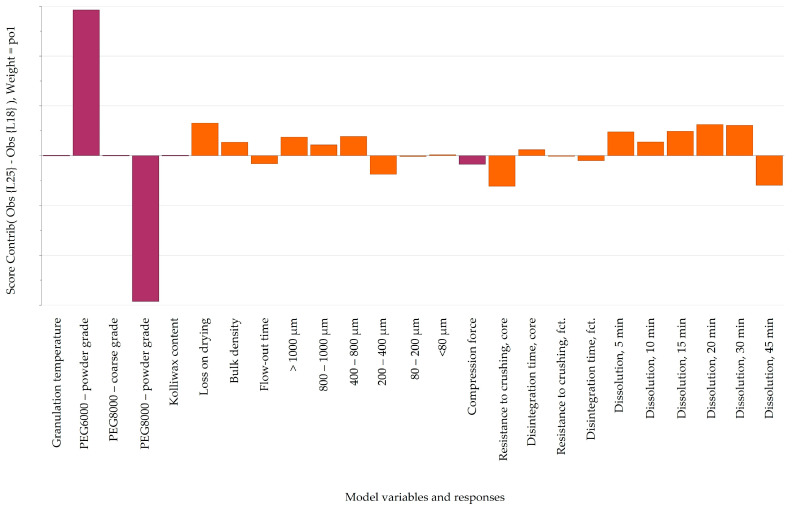
Pairwise comparison of laboratory-scale batches manufactured with Macrogol 6000P (L25) and Macrogol 8000P (L18). Purple bars indicate variable settings, while orange bars show selected responses.

**Figure 4 pharmaceutics-16-00391-f004:**
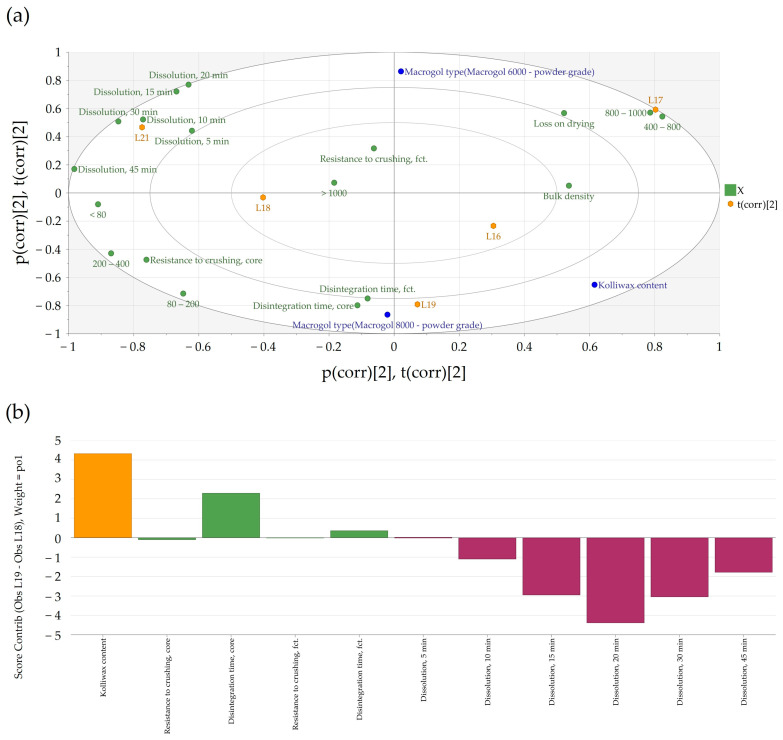
(**a**) Bi-plot of PCA-X modeling of the batches intended for the investigation of the impact of Kolliwax content. The blue circles indicate the manufacturing variables, and the green circles indicate the in-process control responses, while the orange hexagons denote the manufactured batches. (**b**) Contribution plot comparing the L18–L19 laboratory-scale batches (L18—1.33% Kolliwax HCO; L19—2.00% Kolliwax HCO). The Kolliwax content correlates with an increased disintegration time; this, in turn, influences the release of Clopidogrel from the pharmaceutical formulation. The orange bar indicates the Kolliwax content as a factor setting, and the green bars indicate the influences in-process control responses, while the purple bars denote the dissolution marks at different time points.

**Figure 5 pharmaceutics-16-00391-f005:**
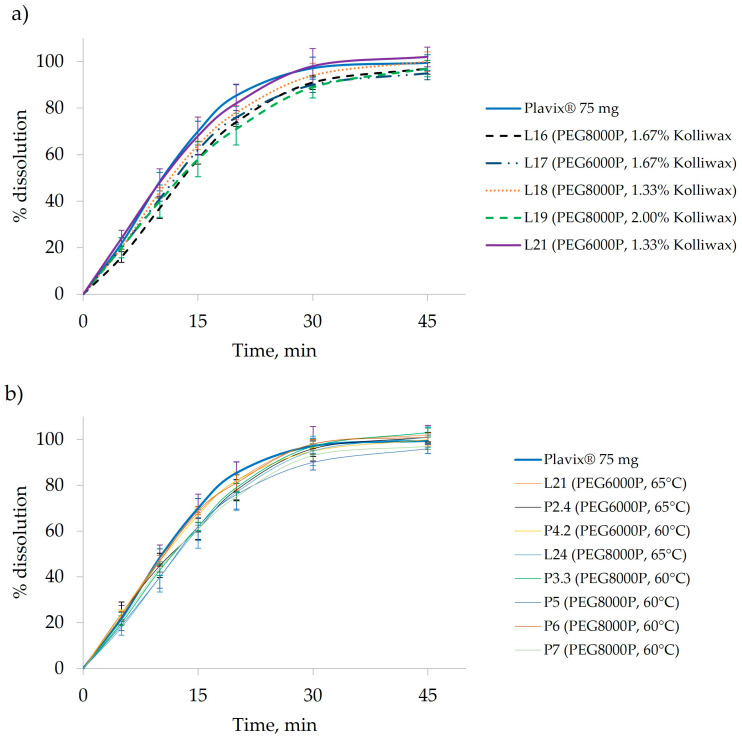
(**a**) Effect of Kolliwax HCO on Clopidogrel dissolution. Comparative dissolution profiles of L16–L21 compared to the reference product, Plavix 75 mg film-coated tablets. *f*2 values compared to Plavix 75 mg are as follows: L16-*f*2 = 49, L17-*f*2 = 57, L18-*f*2 = 64, L19-*f2* = 50, and L21-*f*2 = 82. (**b**) Comparison of dissolution between laboratory- and pilot-scale batches: L21-*f*2 = 82, P2.4-*f*2 = 61, P4.2-*f*2 = 79, L24-*f*2 = 58, P3.3-*f*2 = 62, P4-*f*2 = 56, P5-*f*2 = 81, and P6-*f*2 = 57.

**Figure 6 pharmaceutics-16-00391-f006:**
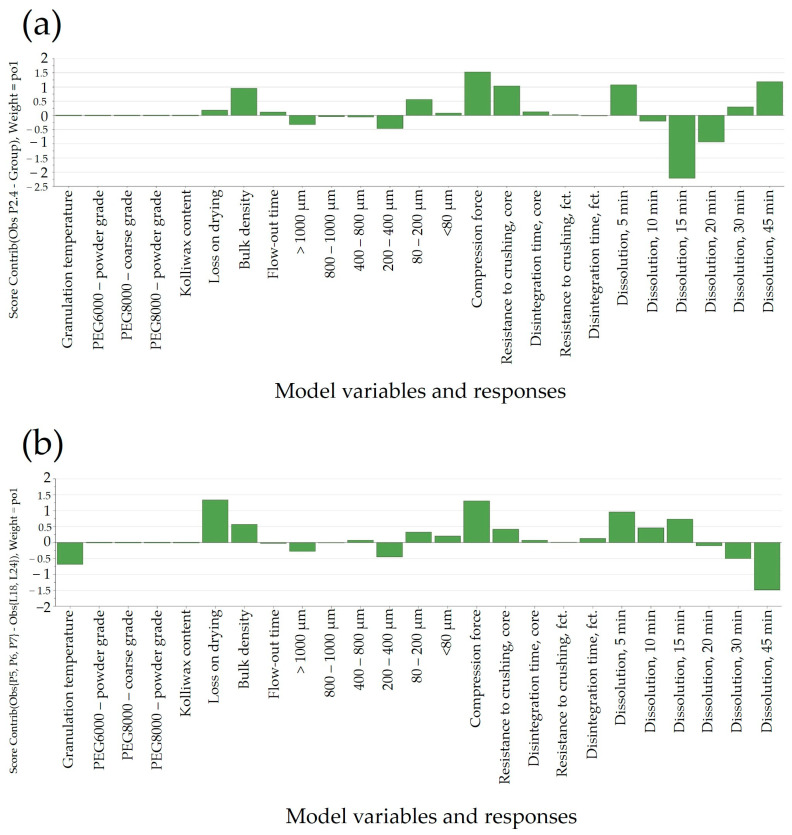
(**a**) Comparison of laboratory-scale batches (groups L21–L23 and L25) with a pilot-scale batch (P2.4) manufactured with PEG6000P and (**b**) comparison of laboratory-scale batches (L18 and L24) with a pilot-scale batch (P5–7) manufactured with PEG8000P.

**Figure 7 pharmaceutics-16-00391-f007:**
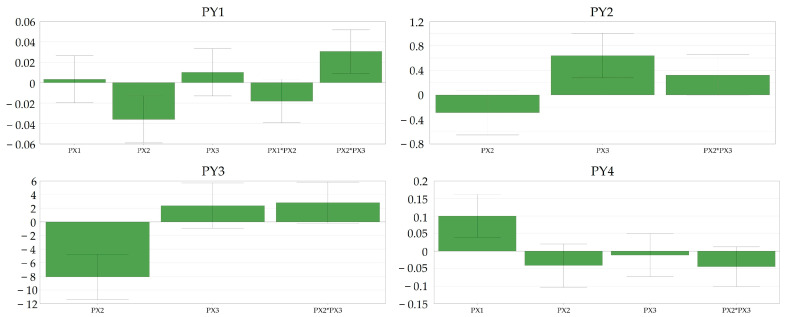
Coefficient plots indicating the direction of magnitude of the defined factors for tablet compression in selected responses.

**Figure 8 pharmaceutics-16-00391-f008:**
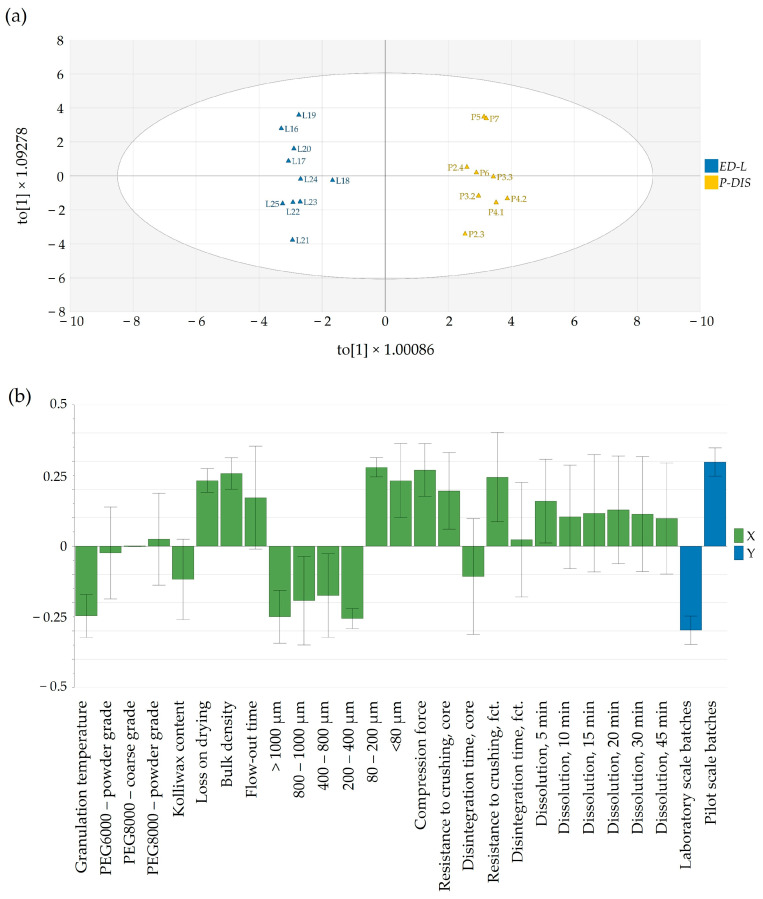
(**a**) Score scatter plot showing the difference between laboratory- and pilot-scale batches (t[1]) and orthogonal variability, capturing differences in dissolution between batches manufactured at the laboratory scale and the pilot scale (to[1]). (**b**) Loading plot of the OPLS-DA model, showing differences between laboratory- and pilot-scale batches.

**Figure 9 pharmaceutics-16-00391-f009:**
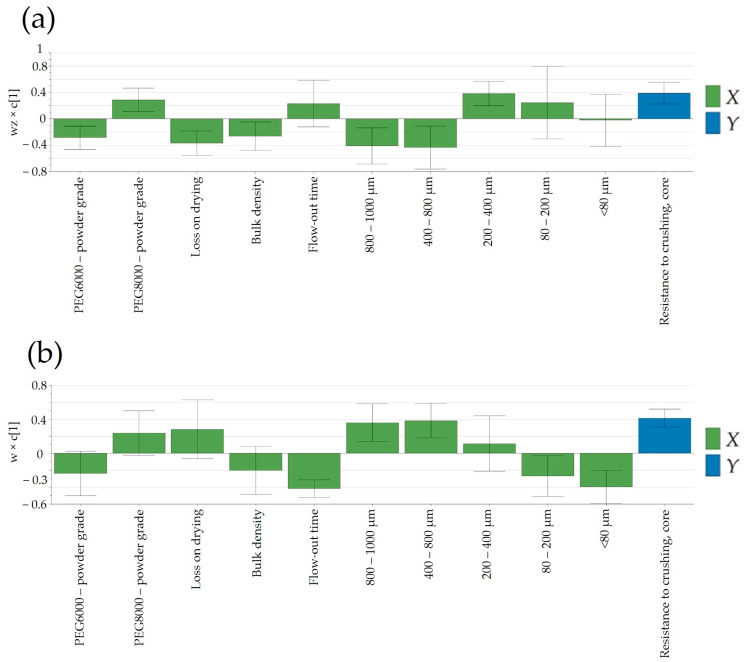
Loading plots of PLS models developed for the prediction of resistance to crushing: (**a**) laboratory scale and (**b**) pilot scale.

**Figure 10 pharmaceutics-16-00391-f010:**
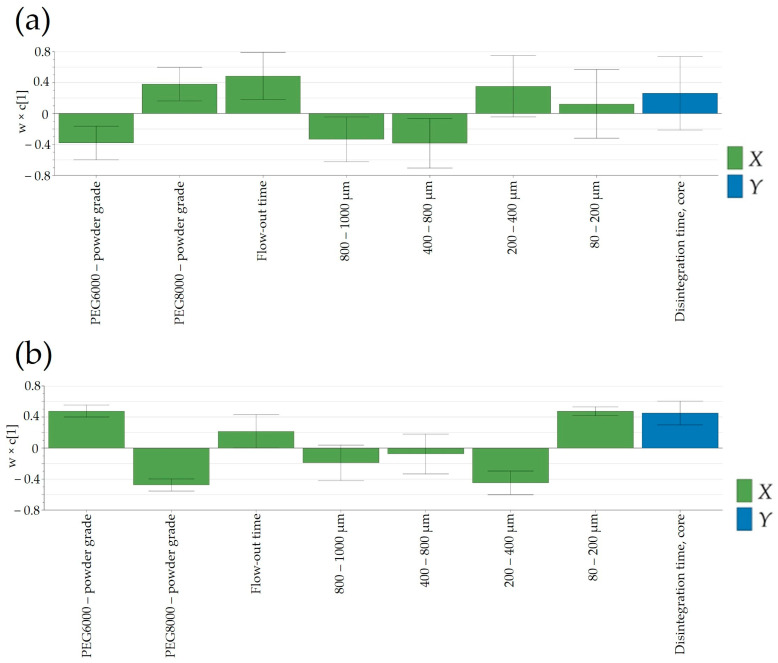
Loading plot of PLS model developed for the prediction of disintegration time at the pilot scale: (**a**) laboratory scale and (**b**) pilot scale.

**Figure 11 pharmaceutics-16-00391-f011:**
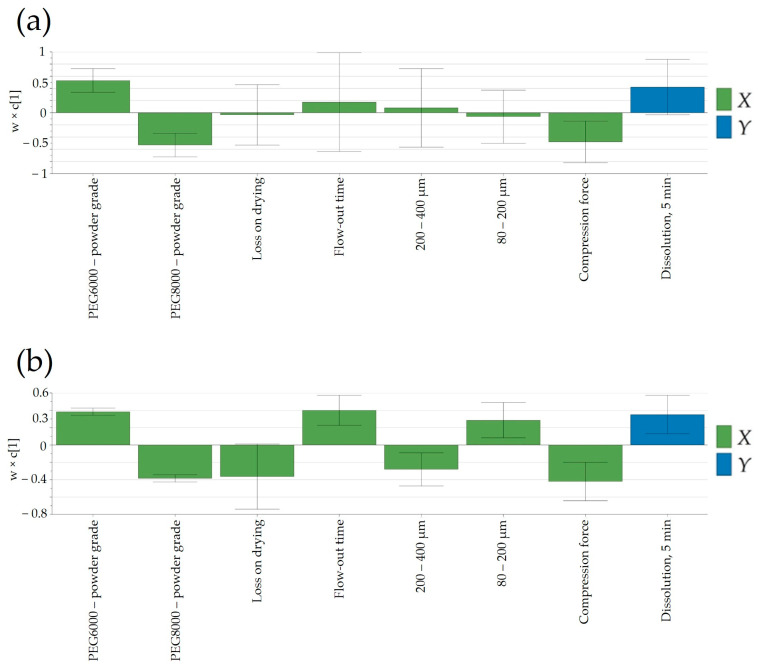
Loading plot of the PLS model developed for the prediction of dissolution: (**a**) laboratory scale and (**b**) pilot scale.

**Table 1 pharmaceutics-16-00391-t001:** Quantitative and qualitative composition of Clopidogrel tablets.

Components	mg/Tablet	%/Tablet (m/m%)	Pharmaceutical Role
Clopidogrel hydrogen sulfate, form II	97.875 *	39.465	API
Mannitol 35	68.925	27.795	Filler
Cellulose microcrystalline type M103D+	31.00	12.50	Filler
Macrogol 6000 or 8000	34.00	13.71	Binder
Low-substituted hydroxypropylcellulose (L-HPC, LH-11)	12.90	5.20	Binder
Hydrogenated castor oil (Kolliwax HCO)	3.30	1.33	Lubricant
Total tablet	248.00	100.00	

* Corresponds to 75 mg of Clopidogrel base.

**Table 2 pharmaceutics-16-00391-t002:** List of factors and responses defined in the experimental design intended for the analysis of CPPs and CMAs.

**FACTORS**					
Factor Name	Abbreviation	Type	Settings		Units
Fluidized bed temperature	LX1	Quantitative	55 to 65		°C
Granulation time	LX2	Quantitative	5 to 30		Min
Macrogol type	LX3	Qualitative	Macrogol 8000–powder grade (PEG 8000P) Macrogol 6000–powder grade (PEG 6000P) Macrogol 8000–coarse grade (PEG 8000C)		-
**RESPONSES**					
Response name	Abbreviation	Min	Target	Max	Units
Bulk density	LY1	0.40	0.50	0.60	g mL^−1^
Flow-out time	LY2	10	25	60	s
PSD (>1000 μm)	LY3a	1.6	2.0	2.4	%
PSD (800–1000 μm)	LY3b	2.4	3.0	3.6	%
PSD (400–800 μm)	LY3c	8.0	10.0	12.0	%
PSD (200–400 μm)	LY3d	20.0	25.0	30.0	%
PSD (80–200 μm)	LY3e	32.0	40.0	48.0	%
PSD (<80 μm)	LY3f	16.0	20.0	24.0	%

PSD—particle size distribution.

**Table 3 pharmaceutics-16-00391-t003:** Factors and responses considered for the evaluation of the tablet compression process.

**FACTORS**					
Factor Name	Abbreviation	Type	Settings		Units
Compression force	PX1	Quantitative	16 to 22		kN
Turret rotation speed	PX2	Quantitative	20 to 80		rpm
Feeder rotation speed	PX3	Quantitative	20 to 80		rpm
**RESPONSES**					
Response name	Abbreviation	Min	Target	Max	Units
Disintegration time	PY1	-	-	900	s
Process capability of tablet mass variation, C_pk_	PY2	1.33	-	-	-
Resistance to crushing	PY3	100	150	180	N
Friability	PY4	-	-	1	%

## Data Availability

Data are contained within the article.

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
