# Peer review of "The Effect of Formulation Variables on the Manufacturability of Clopidogrel Tablets via Fluidized Hot-Melt Granulation—From the Lab Scale to the Pilot Scale"

_pharmaceutics, 2024, doi:10.3390/pharmaceutics16030391_

Round 1

Reviewer 1 Report

Comments and Suggestions for Authors

The manuscript "Study of the effect of formulation variables on the manufacturability of Clopidogrel tablets by fluidized hot-melt granulation - from lab to pilot scale" describes the development of immediate release tablets from granules prepared by fluidized hot melt granulation. The study brings some interesting data and discussion. However, it is very hard to follow, as the writing and the organization of the manuscript is confusing. This is my main concern, which makes difficult a deeper evaluation about its merit to be published. Please, find some additional comments below:

- Introduction is too long and should go direct to the main point of the study; 

- I understood that Clopidogrel was used as a model drug, but at least its uses in the patient treatment as well as its available commercial dosage forms should be stated in the introduction;

- Although a good set of characterization tools was used to evaluate the granules and the tablets, the assay of the drug content in the formulation is missing. Also, it would be of great interest to study the physical state of the drug after the processing steps and if the fast release could be related to an amorphization of the drug;

- Could you explain a little bit further how was carried out the process of lubrification of the granules after the fluid bed step?

- Why pH 2.0 was selected as the dissolution medium?

- There are to many figures in the manuscript and most of them are difficult to follow. Furthermore, some of them are impossible to read (please revise the font size). Please, check also the needing of all tables.

- Results and Discussion section should be significantly condensed. It is too long and difficult to follow. Please, keep the focus on the main data to be highlighted and on those that will add new knowledge to the area.

- Conclusion is too long. Be concise. It looks like an abstract.

- Please revise the use of acronyms in the abstract.

Reviewer 2 Report

Comments and Suggestions for Authors

Comments:

Materials/equipment/methods. you use film coating in your workflow and show LOD measurements. these however do not appear to be detailed in terms of plant used/method.

Figures 3, 7a, 13a. these figures contain very small text that is very difficult to read. I would recommend an edit and increase.

figures 4 and 10. I would recommend that the y axis be fully labelled. I am assuming it is wt% oversize retained on a given sieve? can you also clarify the sieve size. i.e in your supplementary data you use 80-200. Does this actually mean that this is the weight % retained on an 80um sieve. that being the case I would label the x-axis to correspond to the sieve size. i..e pan, 80, 200, 400, 800, 1000 and show the actual points. the frequency distributions are not presented as conventional sieve data layout styles.

figures 8 and 11 on dissolution. It would be valuable to include error/range bars. I am assuming that the dissolution was carried out in replicates, but this is not clear and therefore not fully clear the variation on dissolution of a given preparation.

Tablet crushing strength data and approach. I would suggest/recommend that for this measurement that comparisons are made at a given solid fraction where tablet mass and dimension are accounted for .

reference 37 has a typo.

Supplementary material 1: on the database for the MVDA. Please put in the units of the measurements where applicable. It also appears that you have particle size/sieve data. please detail this as to what it is. it is currently just noted as 800-1000, 400-800 etc. please specify and add units.

Supplementary materials. Bar-Plots. please label the axes with appropriate descriptor.
